# Early Detection of Disease Progression in Metastatic Cancers: Could CTCs Improve RECIST Criteria?

**DOI:** 10.3390/biomedicines12020388

**Published:** 2024-02-07

**Authors:** Valentina Magri, Luca Marino, Gianluigi De Renzi, Michela De Meo, Francesca Salvatori, Dorelsa Buccilli, Vincenzo Bianco, Daniele Santini, Chiara Nicolazzo, Paola Gazzaniga

**Affiliations:** 1Department of Pathology, Oncology and Radiology, Sapienza University of Rome, 00161 Rome, Italy; francesca.salvatori@uniroma1.it (F.S.); dorelsa.buccilli@uniroma1.it (D.B.); vincenzo.bianco@uniroma1.it (V.B.); daniele.santini@uniroma1.it (D.S.); 2Department of Mechanical and Aerospace Engineering, Sapienza University of Rome, 00161 Rome, Italy; luca.marino@uniroma1.it; 3Department of Molecular Medicine, Sapienza University of Rome, 00161 Rome, Italy; gianluigi.derenzi@uniroma1.it (G.D.R.); michela.demeo@uniroma1.it (M.D.M.); chiara.nicolazzo@uniroma1.it (C.N.); paola.gazzaniga@uniroma1.it (P.G.)

**Keywords:** metastatic cancers, circulating tumor cells, RECIST criteria, disease progression, personalized oncology

## Abstract

Early detection of disease progression is a crucial issue in the management of cancer patients, especially in metastatic settings. Currently, treatment selection mostly relies on criteria based on radiologic evaluations (RECIST). The aim of the present retrospective study is to evaluate the potential inclusion of circulating tumor cells (CTCs) in hybrid criteria. CTC counts from a total of 160 patients with different metastatic tumors were analyzed for this purpose. In our cohort, 73 patients were affected by breast cancer, 69 by colorectal cancer and 18 by prostate cancer. PFS and OS were evaluated according to the corresponding prediction of disease progression by CTCs and RECIST criteria. In breast cancer, CTC-I has an important impact on the progression-free survival (PFS) and overall survival (OS) values. When CTC-I predicted earlier than RECIST-I, the disease progression, the PFS and OS were shorter with respect to the opposite case. In particular, PFS was 11 (5–16) vs. 34 (23–45)—with *p* < 0.001—and OS was 80 (22–138) vs. 116 (43–189), *p* = 0.33. The results suggest a promising role of CTCs as complementary information which could significantly improve the clinical outcomes, and they encourage consideration of future trials to evaluate new hybrid criteria, particularly for patients with breast cancer.

## 1. Introduction

Making therapeutic decisions for cancer patients is often a hard task for oncologists, especially in a metastatic setting of disease. Metastatic cancers are generally non-curable diseases, and the treatment goal is mainly palliative. In selected groups of patients, however, long-term survival is possible, and, in these scenarios, a patient’s death could be related to different diseases, sometimes cancer-independent [1,2,3]. Cancer treatment is a dynamic process that must be tailored to the patient and depends on multidimensional parameters linked to objective and subjective issues. The selection of the best and most effective therapies, with optimal timing and minimum toxicities, is crucial to improve the quality and the extent of life in these conditions. Fundamental in the treatment decisions are patients’ will, preference and quality of life [1].

A comprehensive evaluation of the prognosis and of the balance between efficacy and toxicity should guide the therapeutic choices. Once the oncologic treatment has started, constant monitoring is required. The time extent of the treatment depends on several factors. Disease progression, withdrawal of patient consent or occurrence of unacceptable loss of clinical benefit are reasons to stop and, eventually, change the pharmacological therapy.

The tumor response to the current adopted therapeutic protocols is performed every 3–4 cycles of treatment. Oncological history, clinical course, laboratory data (serology, organ and tumor specific marker patterns) and imaging evaluation are all considered to assess regression, stability or progression of the disease. The treatment efficacy is evaluated in terms of measurable endpoints such as progression-free survival (PFS) and response rate (RR). Even if these surrogates of overall survival (OS)—in particular the RR—are suboptimal, they are routinely used as end points in clinical trials and in daily clinical practice. To date, a recent meta-analysis reported that 82% of the correlations between oncologic surrogate markers (PFS and OS) are not significant, and it is therefore apparent that better markers of survival are required [4].

Early detection of disease progression is crucial to switch to following therapeutic lines or—in some cases—to administer a concomitant palliative care treatment, which can provide significant improvements in both quality of life and OS [5]. Among the several types of data available to oncologists to assess the status of the disease, that offered by radiologic tools is one of the most significant. In fact, radiologic disease evaluation based on the imaging-based Response Evaluation Criteria in Solid Tumors (RECIST v1.1) represents the best approach to evaluating the oncologic patient state during treatment to date. The RECIST criteria are based on the comparison—with reference to the baseline conditions—of the size of cancer lesions as obtained at each disease evaluation [6].

Even if routinely adopted in clinical practice, RECIST criteria show several limitations. First, they depend on radiologist estimation of cancer dimensions and on the specific imaging technique adopted. Moreover, an underestimation of disease progression is possible as RECIST do not consider the biological state of cancer clones. In fact, clonal expansion frequently leads to rapid growth following treatment. Typically, drug-resistant clones exhibit such rapid expansion, and the progression of the disease is closely linked to this phenomenon.

Even if nowadays a personalized treatment is sought, oncologists often decide to continue or to stop a specific cancer treatment without complete knowledge of the state of the tumor biology at the time of that precise evaluation, with possible important consequences on patient well-being, quality of life and survival. 

In this context, liquid biopsy represents a precious tool to optimize patient’s evaluation, providing important information concerning circulating tumor analytes and having several advantages, including repeatability, tolerability, rapidity and cost-effectiveness. Circulating tumor cells (CTCs) are rare events; they are released from the primary tumor in the bloodstream—where few survive—and successively undergo an epithelial-to-mesenchymal transition (EMT) in order to complete metastatic cascade [7]. Since anti-cancer therapies could act as selective pressure for tumor cells and can influence clonal expansion, liquid biopsies repeated during treatment are able to unveil the dynamic changes that occur in tumors during treatment, providing a wider view on the biological state of cancer clones [8,9].

The FDA-approved CellSearch^®^ system is a semi-automated system performing immunomagnetic EpCAM-based CTC isolation, and it is validated for the enumeration of CTCs in metastatic breast, colorectal and prostatic cancers. Several studies have shown that CTCs provide prognostic information in large patient cohorts [10,11]. 

However, despite years of expectation, CTCs have failed to overcome their limitations in a manner necessary to be included in routine clinical practice.

The presence of CTCs might be indicative of the onset of resistance phenomena, thus allowing the modification of therapeutic strategy long before evidence of clinical or radiological progression. The aim of the present study is to investigate the potential value of serial CTC enumerations compared to RECIST v1.1 criteria in metastatic cancer patients during treatments.

## 2. Materials and Methods

The present retrospective analysis aims to elucidate the potential value of integrating conventional RECIST v1.1 criteria and CTC serial enumeration in a cohort of metastatic cancer patients during treatments. A query from our institutional medical record database was performed to identify patients affected by histologically confirmed breast or colorectal or prostate cancer who underwent a liquid biopsy during any metastatic cancer treatment from June 2010 to December 2012. The set of patients considered was composed of 73 subjects with breast cancer, 69 with colorectal cancer and 16 with prostate cancer. Out of the 73 patients affected by breast cancer, 30 were in I-line therapy and 43 were in II- or III-line therapy. All the patients with colorectal and prostate cancer were in I-line therapy.

The inclusion criteria were breast, colorectal or prostate cancer patients with measurable metastatic disease undergoing any line of treatment for metastatic disease. CellSearch^®^ analysis was performed at each radiological evaluation; RECIST 1.1 assessment was performed and follow-up data are available. A sample of 7.5 mL of peripherical blood was drawn for each patient at the same time of each radiological evaluation, approximately every 3 months or at PD.

The follow-up, defined as the median (IQR25–IQR75) of the period ranging from the date of diagnosis for patients enrolled to the date of death, was 94 (41.7–253) months. Patient data were collected using Excel 2011 (version 14.0, Microsoft Corporation, Redmont, WA, USA).

### 2.1. CTC Enumeration

The CTC enumeration was carried out with the CellSearch^®^ device (Menarini Silicon Biosystems, Castel Maggiore, Italy), which is a part of our group’s equipment and is available in the Liquid Biopsy Unit within the Department of Molecular Medicine. The CellSearch^®^ system utilizes a CellSearch^®^ Epithelial Cell Kit (Menarini Silicon Biosystems), which allows CTC enrichment through an anti-EpCAM-antibody-coated ferrofluid reagent, followed by staining for cytokeratins (CK), 4′-6-Diamidino-2-phenylindole (DAPI) and CD45. The peripheral blood was gathered in a CellSave tube containing EDTA and a cell fixative at room temperature and processed within 72 h. A recorded event was assumed to be a CTC when it had round-to-oval morphology, a visible nucleus, positive staining for CK and negative staining for CD45.

### 2.2. Criteria of Response Evaluation

Therapeutic choices were exclusively made according to RECIST v1.1 criteria [6]. CTC trajectories were recorded, but they had no role in clinicians’ decisions on the type and timing of treatments. Information on the CTC count was adopted in terms of a closer patient follow-up with a particular type of care upon symptom onset. Moreover, patients were not informed regarding CTC progression. In this study, two indexes were introduced to evaluate the possible role of CTC count in hybrid criteria. An index RECIST-I was defined as positive when the corresponding RECIST criteria predicted progression disease (PD), and an index CTC-I was defined as predicting PD according to values of CTCs above the approved cut-offs. The PD was established when CTC count increased with respect to the previous value above the proper, cancer-specific cut-off. In the particular case of a baseline value above the cut-off, any increase of CTC counts corresponded to PD. Details of the RECIST and CTC criteria are summarized in Table 1.

### 2.3. Statistical Analysis

For the present study, PFS and OS were the clinical endpoints. Categorical variables were reported as a frequency distribution, whereas continuous variables were described with the median with a 95% confidence interval (CI). Survival curves were represented through a Kaplan–Meier analysis and the differences between the groups were evaluated with a log-rank test. A *p*-value less than 0.05 was considered statistically significant. All statistical tests were 2-sided. Statistical analyses were carried out using SPSS Statistics software version 25.0 (IBM Corp., Armonk, NY, USA).

## 3. Results

The three cohorts of patients analyzed (breast, colorectal and prostate cancer) were different in terms of numerosity, PFS and OS. In particular, in the breast cancer cohort (73 patients), the median PFS was 23 months (95% CI = 14–33) and OS was 103 months (95% CI = 80–120). For colon cancer (69 patients), the median PFS was 16 months (95% CI = 13–18) and the median OS was 38 months (95% CI = 29–46). Finally, for prostate cancer (18 patients), the median PFS was 9.8 months (95% CI = 1.2–18), while the median OS was 110 months (95% CI = 84–137), as shown in Table 2. The PFS and OS values evaluated according to the prediction of either CTC-I or RECIST-I are shown in Table 3.

For breast cancer, CTC-I was found to be significantly associated with PFS and OS. The PFS value was 29 months (20–38) for CTC-I = No-PD and 16 months (5–27) for CTC-I = PD (*p*-value = 0.1). According to RECIST-I, PFS was 20 months (2–37) for RECIST-I = No-PD and 24 months (13–35) for RECIST-I = PD (*p*-value = 0.33). The median OS was 116 months (56–174) for CTC-I = No-PD and 80 months (36–124) for CTC-I = PD (*p*-value = 0.05), while the median OS was 108 months (56–159) for RECIST-I = No-PD and 103 months (77–128) for RECIST-I = PD (*p*-value = 0.64).

For colon cancer, the median PFS was 16 months (13–18) for CTC-I = No-PD and 14 months (10–17) for CTC-I = PD (*p*-value = 0.85). The RECIST-I analysis provided the same PFS value (16 months) (12–19) for both RECIST-I. The median OS was 38 months (25–50) for CTC-I = No-PD; 34 months (22–45) for CTC-I = PD (*p*-value =0.76); 41 months (22–59) for RECIST-I = No-PD and 34 months (28–39) for RECIST-I = PD (*p*-value = 0.17).

For prostate cancer, no differences were observed for PFS according to the two progression indexes. The median PFS was 14 months (3/4–24/25) for CTC-I/RECIST-I = No-PD and 6 months (3–9) for CTC-I/RECIST-I = PD (*p*-value = 0.4/0.2). The median OS was 133 months (97–169) for CTC-I = No-PD; 75 months (72–78) for CTC-I = PD (*p*-value = 0.17); 122 months (88–150) for RECIST-I = No-PD and 75 months (60–140) for RECIST-I = PD (*p*-value = 0.17).

PFS and OS were evaluated according to the correspondence between the two prediction indexes—CTC-I and RECIST-I—for the three different pathologies, and the values are reported in Table 4a–c.

For breast cancer (Table 4a), when CTC-I and RECIST-I did not agree, PFS and OS were significantly shorter according to the CTC-I prediction of disease progression. In particular, the median PFS was 10 months (5–17) when CTC-I = PD and 27 months (17–38) when CTC-I = No PD, with a *p*-value = 0.04. The median OS was 49 months (6–92) when CTC-I = PD and 146 months (61–232) when CTC-I = No PD, with a *p*-value = 0.05.

When CTC-I and RECIST-I were found to be concordant, both PFS and OS showed shorter values when the two criteria predicted PD. PFS was 29 months (7–50) when CTC-I/RECIST-I = No PD and 19 (2–37) when CTC-I/RECIST-I = PD, with a *p*-value = 0.21. The median OS was 108 months (22–194)) when CTC-I/RECIST-I = No PD and 80 months (39–121) when CTC-I/RECIST-I = PD, with a *p*-value = 0.27. Analogous results (not reported) were obtained for the two groups of patients, divided according to the I-line therapy or a successive II- or III-line therapy.

For colon cancer (Table 4b), when CTC-I and RECIST-I were discordant, PFS and OS were analogously shorter according to the CTC-I prediction of disease progression. In particular, the median PFS was 12 months (11–14) when CTC-I = PD and 16 months (12–19) when CTC-I = No PD, with a *p*-value = 0.2. The median OS was 25 months (18–32) when CTC-I = PD and 34 months (27–40) when CTC-I = No PD, with a *p*-value = 0.5. In this peculiar case, the longer values of PFS and OS, though a PD is predicted by RECIST-I, could be related to the low CTCs (No PD), which is a well-known index of better prognosis [11].

When CTC-I and RECIST-I were concordant, the median PFS was 16 months (12–19) when CTC-I/RECIST-I = No PD and 15 months (11–19) when CTC-I/RECIST-I = PD, with a *p*-value = 0.7. The median OS was 44 months (16–71) when CTC-I/RECIST-I = No PD and 34 months (23–45) when CTC-I/RECIST-I = PD, with a *p*-value = 0.4. The agreement of the two indexes confirms the important role of CTCs in the clinical course of the diseases.

For prostate cancer (Table 4c), when CTC-I and RECIST-I were discordant, the median PFS and OS had similar values. In particular, the median PFS was 6.4 months (4–8) when CTC-I = PD and 6.6 months (3–10) when CTC-I = No PD, with a *p*-value = 0.5. The median OS was 74 months (0–170) when CTC-I = PD and 110 months (0–280) when CTC-I = No PD, with a *p*-value = 0.51.

When CTC-I and RECIST-I were concordant, the median PFS was 18 months (7–29) when CTC-I/RECIST-I = No PD and 6.2 months (0–14) when CTC-I/RECIST-I = PD, with a *p*-value = 0.2. The median OS was 133 months (100–165) when CTC-I/RECIST-I = No PD and 75 months (31–119) when CTC-I/RECIST-I = PD, with a *p*-value = 0.16.

In Table 5, the values of PFS and OS are reported in cases where PD occurred. The data are divided into two groups according to the first index which predicted the PD (progression index, PI).

For breast cancer, PFS and OS were significantly shorter when CTC-I predicted PD before RECIST-I (PI = 1). In detail, the median PFS was 11 months (5–16) when the PI = 1 and 34 months (23–45) when the PI = 2 (RECIST-I predicts PD before CTC-I), with a *p*-value < 0.001. The median OS was 80 months (22–138) when the PI = 1 and 116 months (44–189) when PI = 2, with a *p*-value = 0.33.

For colon cancer, the median PFS was 17 months (7–27) when the PI = 1 and 15 months (12–17) when the PI = 2, with a *p*-value = 0.63. The median OS was 34 months (28–40) when the PI = 1 and 38 months (23–53) when PI = 2, with a *p*-value = 0.98.

For prostate cancer, the median PFS was 6.4 months (6–7) when the PI = 1 and 4.5 months (3–6) when the PI = 2, with a *p*-value = 0.71. The median OS was 26 months (21–31) when the PI = 1 and 13.7 months (2–24) when PI = 2, with a *p*-value = 0.81.

The effect of treatment on CTC counts and CTC-I and RECIST-I was investigated only for breast cancer. The cohort of patients studied with colorectal and prostate cancer was under a single therapy line (chemotherapy + bevacizumab and endocrine therapy, respectively).

For breast cancer, the patients were divided into three groups according to the different therapy adopted: chemotherapy (CHT), hormone therapy (HT) and the presence of target therapy (TT). For breast cancer, no significant differences were found between the HT and TT groups. In particular, the mean CTC count was 2 ± 3 in the HT group and 4.1 ± 12.6 in the TT group. The CHT group of patients presented a higher value of CTC count (55 ± 192). The patients treated with CHT presented a more aggressive disease with respect to the more indolent nature of the Luminal cancer subtype commonly treated with HT. This issue could be a possible reason for the largest number of CTCs found. To the best of the authors’ knowledge, this aspect was not investigated in the pertinent literature. Table 6 shows the details of the CTC counts. The analyses of PFS and OS, according to the progression index PI, which takes into account the specific treatment adopted in the breast cancer, are reported in Table 7. The patients under HT and TT had lower values of PFS and OS when the index PI = 1 compared to the whole set of patients (Table 5). The group with CHT treatment did not record differences in the value of OS (Table 7).

## 4. Discussion

We investigated the possible role of CTC enumeration as a tool to integrate the conventional RECIST criteria to achieve an early detection of disease progression in metastatic breast, colorectal and prostate cancers.

The prognostic role of baseline CTC enumeration in these tumor types has been well established in several clinical trials. Specifically, a CTC count of ≥5 cells per 7.5 mL blood detected by the CellSearch^®^ system is an independent predictive factor of worse progression-free survival (PFS) and overall survival (OS) in breast and prostate cancer, while in metastatic CRC, the prognostic cut-off was set to ≥3 cells per 7.5 mL [12,13]. This evidence guided the CellSearch clinical approval by the FDA as the gold standard to detect and enumerate CTCs in these cancer types. Moreover, the superiority of a biological CTC approach over the traditional imaging method as an indicator of disease status has already been demonstrated. CTCs turned out to be better predictors of OS than disease changes evaluated with traditional imaging in metastatic breast cancer patients [14].

Despite the strong biological rationale assuming that CTCs might better reflect the tumor state, few interventional clinical trials have been specifically designed to demonstrate that CTCs can improve cancer treatment decisions [15].

Among these, the STIC CTC trial, a multicenter, prospective, phase III trial, demonstrated that in breast cancer, the CTC-driven treatment decision arm was non-inferior to the clinician’s choice arm in terms of 2 years PFS. In the STIC CTC trail, 761 metastatic breast cancer patients were randomized into either a clinician’s choice arm, where the decision to administer hormone therapy (HT) or chemotherapy (CT) was made clinically before CTC results were performed, or a treatment CTC-driven arm, where HT or CT were administered if CTCs were <5/7.5 mL or ≥5/7.5 mL, respectively. Interestingly, among 202 patients with discordant features (Clin_low_/CTC^high^ or Clin^High^/CTC_low_), only the 189 patients with Clin_low_/CTC^high^ features reached statistically better PFS and OS receiving CT (CTC-driven decision) rather than ET (clinical choice) [15]. Unfortunately, this clinical trial was designed before the introduction of Cyclin-Dependent Kinase 4/6 inhibitors, which have proven their significant advantage in terms of PFS and partially OS, thus delaying the need to introduce CT in luminal breast cancer [2,16,17].

Despite this limitation, STIC CTC is a milestone trial, definitely demonstrating that CTC-driven treatment decision-making is not inferior to oncologists’ choices based on clinical estimation of cancer risk. However, doubts may persist regarding the rationale of using CTCs without their molecular characterization to select CT or ET treatment options.

In light of what has been said so far, our study could be a trailblazer, since it proposes for the first time—to the best of the authors’ knowledge—an integration between the conventional RECIST criteria and CTC serial enumeration, namely a “CTC-RECIST” evaluation. Despite being a retrospective analysis on a limited number of patients, the intent is to spark a new topic of discussion, specifically to clarify whether a more biological approach can improve the clinical evaluation of the state of tumor disease.

Of the metastatic cancer patients enrolled in this study, the most interesting results were obtained for breast cancer, where a PD progression as predicted by the CTC-I was associated—with statistical significance—with a shorter OS with respect to the case of CTC-I not predictive of PD. Even more interesting, when RECIST-I and CTC-I were discordant in PD prediction, CTC-I better predicts PD compared to RECIST-I. This outcome could be related to a delay in the therapy’s change as RECIST failed to indicate a PD, possibly supporting the added value of CTC enumeration as a predictive biomarker of disease progression. Consistently, survival was longer when only the RECIST-I predicted PD and, as a consequence, guided the therapy’s change early. Notably, the same behavior is observed if the set of patients with breast cancer is split into a group of patients under an I-line therapy and a group of patients under an II- or III-line therapy. This result confirms that the prognostic role of the CTC is independent of the particular therapy line adopted.

Our analysis also supports the importance of CTCs in PD prediction. In 11 breast cancer patients, CTC-I predicted PD before RECIST-I with a deep impact on PFS, which was significantly shorter (11 vs. 34 months *p* < 0.001). This result unveils the need for a therapy’s switch at the proper time for a subgroup of breast cancer patients. Moreover, CTCs appear to be more sensitive predictive biomarkers of treatment response than the RECIST criteria.

Notably, 45% of these patients had no PD predicted by RECIST, while in 55%, the mean lag time between CTC-I and RECIST-I PD prediction was 4.5 months. Remarkably, in this small subset of patients, tumor markers (CEA, Ca15.3) showed highly oscillating values at each check, confirming their limited utility in the management of patients’ therapy in the context of a possible PD [18].

In metastatic colon cancer patients, although we failed to reach significant results, trends in survival outcomes were found to be concordant with the results obtained in the breast cancer group. Evidence has been provided by different research groups concerning the limitations of CTC enumeration in mCRC [19,20]. In the present study, the unclear role of colorectal CTC emerged again, as recently demonstrated by Magri et al. [21].

It is worthwhile to discuss some limitations of the present study. First, we observe that the sample size of the prostate patients enrolled is poor and no significant conclusion can be obtained for this specific tumor. Second, the collection of blood samples was obtained at the same time for all patients, regardless of the specific treatment on course. This procedure, largely adopted in clinical practice [22], can introduce a bias in the results. In fact, the fluctuation of CTC content after treatment is currently unknown, and assuming that treatment is effective, the number of CTCs will decrease. The numerical statistics of CTCs in patients with different treatment strategies could be affected by the sampling at the same time interval. Moreover, the biological subtype of the cancer can also influence the CTC count, with a higher rate of CTCs released in highly aggressive cancer phenotypes. A further limitation is represented by the blood sample withdrawal. The study was carried out with one single sample obtained upon admission to the day hospital. Compared with taking all of the 7.5 mL of the blood sample at a time, a collection—of the same amount—obtained at multiple times could better ensure the accuracy of CTC counting. In fact, a recent study [23] reports on clinical evidence of the circadian rhythm role in the entire process of tumorigenesis. It is argued that future clinical trials will plan a sample strategy that takes into account the therapy regime and the sampling techniques.

We observe that, in order to improve RECIST criteria, the use of circulating tumor DNA (ctDNA) is also currently under consideration. Occurring soon after the beginning of treatment, ctDNA variations can potentially predict responses to therapy weeks before any evidence of changes in tumor size [24]. Gouda et al. (2022) recently reported an analysis of serial blood samples collected at baseline, mid-treatment and at the time of restaging from patients with different tumor types [25]. The results of this study highlight that higher ctDNA detection rates can be observed in patients with disease progression at first restaging compared to patients with stable disease, partial response or complete response [b]. Despite limitations, the authors reported an association between the detection of ctDNA and the response to therapy, supporting the potential of ctDNA analysis as a valid alternative to radiologic imaging in evaluating treatment outcomes [25]. Although growing evidence seems to confirm the role of ctDNA in the assessment of treatment response, the absence of methodological standardization, together with the lack of a globally accepted definition of ctDNA response and progression, critically hampers the implementation of ctDNA response criteria in clinical practice. Further investigations are also needed to better define which source is more suitable for ctDNA analysis, the clinical setting for this application and the timing of the assessment [24].

## 5. Conclusions

We propose here to unify CTC enumeration and RECIST criteria in a new “hybrid criteria” to achieve early detection of disease progression in metastatic breast cancer. Even if this study is strongly limited by the small sample size, it represents a proof of concept aimed at unveiling the biological state of cancer in real time at each disease evaluation, possibly indicating the most appropriate time to change treatment. This integration between a biological and a clinical parameter sounds closer to the perspective of personalized oncology. For this purpose, future trials should aim to integrate RECIST and CTC criteria through CTC molecular and genetic characterization. In breast cancer, for example, the enumeration and characterization of CTCs for ER, PR, HER2, PI3K, BRCA, etc. is supposed to properly identify not only the timing of treatment changes, but, more importantly, to select the most appropriate treatment type.

## Figures and Tables

**Table 1 biomedicines-12-00388-t001:** Summary of criteria for response evaluation by standard RECIST v1.1 and CTC criteria.

	RECIST v1.1 Criteria	CTC Criteria
Progression disease (PD)	≥20% increase in the sum of diameters of target lesions.	Increase of CTC number above a cut-off of 5 CTCs/7.5 mL in case of breast and prostate cancer, 3 CTCs/7.5 mL in case of colorectal cancer. If baseline CTCs is above the cut-off, any increase of CTCs.
Stable diasease (SD)	Neither sufficient reduction to qualify for PR nor sufficient increase to qualify for PD.	No changes in CTC number or CTCs persistently under cut-off.
Partial response (PR)	≥30% decrease in the sum of diameters of target lesions.	Decrease of CTC number but remaining above cut-off limits.
Complete response (CR)	Disappearance of all target lesions.	Decrease of CTC number to undetectable value (0) or under cut-off limits.

**Table 2 biomedicines-12-00388-t002:** PFS (months) and OS (months) for the studied patients. Values in months.

	Breast Cancer	Colon Cancer	Prostate Cancer
N. of patients	73	69	18
PFS	23 (14–33)	16 (13–18)	9.8 (1.2–18)
OS	103.3 (80–120)	38 (29–46)	110 (84–137)

PFS = progression-free survival, OS = overall survival.

**Table 3 biomedicines-12-00388-t003:** PFS (months) and OS (months) according to the progression disease indexes CTC-I, RECIST-I.

	Breast Cancer		Colon Cancer		Prostate Cancer	
	PFS	OS	PFS	OS	PFS	OS
CTC-I						
No-PD	29 (20–38)	116 (57–174)	16 (13–18)	38 (25–50)	14 (3–24)	133 (97–169)
PD	16 (5–27)	80 (36–124)	14 (10–17)	34 (22–45)	6 (3–9)	75 (72–78)
*p*-value	0.1	0.05	0.85	0.76	0.4	0.17
RECIST-I						
No-PD	20 (2–37)	108 (56–159)	16 (12–19)	41 (22–59)	14 (4–25)	122 (88–150)
PD	24 (13–35)	103 (77–128)	16 (12–19)	34 (28–39)	6 (3–9)	75 (60–140)
*p*-value	0.33	0.67	0.45	0.17	0.22	0.11

PFS = progression-free survival, OS = overall survival, PD = progression disease, CTC-I = circulating tumor cells index, RECIST-I = radiological response evaluation index.

**Table 4 biomedicines-12-00388-t004:** PFS (months) and OS (months) according to correspondence between RECIST-I and CTC-I.

a. Breast Cancer			
CTC-I	RECIST-I	PFS	OS
PD	No PD	10 (5–17)	49 (6–92)
No PD	PD	27 (17–38)	146 (61–232)
		*p*-value = 0.04	*p*-value = 0.05
No PD	No PD	29 (7–50)	108 (22–194)
PD	PD	19 (2–37)	80 (39–121)
		*p*-value = 0.21	*p*-value = 0.27
**b. Colon Cancer**			
**CTC-I**	**RECIST-I**	**PFS**	**OS**
PD	No PD	12 (11–14)	25 (18–32)
No PD	PD	16 (12–19)	34 (27–40)
		*p*-value = 0.2	*p*-value = 0.5
No PD	No PD	16 (12–19)	44 (16–71)
PD	PD	15 (11–19)	34 (23–45)
		*p*-value = 0.7	*p*-value = 0.4
**c. Prostate Cancer**			
**CTC-I**	**RECIST-I**	**PFS**	**OS**
PD	No PD	6.4 (4–8)	74 (0–170)
No PD	PD	6.6 (3–10)	110 (0–280)
		*p*-value = 0.5	*p*-value = 0.51
No PD	No PD	18 (7–29)	133 (100–165)
PD	PD	6.2 (0–14)	75 (31–119)
*p*-value		*p*-value = 0.2	*p*-value = 0.16

PFS = progression-free survival, OS = overall survival, PD = progression disease, CTC-I = circulating tumor cells index, RECIST-I = radiological response evaluation index.

**Table 5 biomedicines-12-00388-t005:** PFS (months) and OS (months) according to progression index PI (1 = CTC-I before RECIST-I, 2 = RECIST-I before CTC-I).

Cancer Location	PI (N. pts)	PFS		OS	
Breast	1 (11)	11 (5–16)	*p*-value < 0.001	80 (22–138)	*p*-value = 0.33
	2 (32)	34 (23–45)		116 (43–189)	
Colon	1 (16)	17 (7–27)	*p*-value = 0.63	34 (28–40)	*p*-value = 0.98
	2 (26)	15 (12–17)		38 (23–53)	
Prostate	1 (3)	6.4 (6–7)	*p*-value = 0.71	26.2 (21–31)	*p*-value = 0.81
	2 (3)	4.5 (3–6)		13.7 (2–24)	

PFS = progression-free survival, OS = overall survival, CTC-I = circulating tumor cells index, RECIST-I = radiological response evaluation index, PI = progression index.

**Table 6 biomedicines-12-00388-t006:** CTC count—mean value ± standard deviation—according to the specific treatment for breast cancer.

Treatment (N. pts)	CTC Count
CHT (20)	55 ± 192
HT (26)	2 ± 3
TT (18)	4.1 ± 12.6

CHT = chemotherapy, HT = hormone therapy, TT = target therapy.

**Table 7 biomedicines-12-00388-t007:** PFS (months) and OS (months) according to progression index PI (1 = CTC-I before RECIST-I, 2 = RECIST-I before CTC-I) for different treatments for breast cancer.

Treatment	PI (N. pts)	PFS		OS	
CHT	1 (4)	11 (4–19)	*p*-value = 0.2	66 (31–103)	*p*-value = 0.75
	2 (12)	24 (4–43)		63 (41–84)	
HT	1 (4)	7 (0–18)	*p*-value = 0.024	104 (15–193)	*p*-value = 0.87
	2 (14)	31 (16–45)		146 (83–210)	
TT	1 (3)	7 (0.6–13)	*p*-value = 0.046	49 (4–94)	*p*-value = 0.15
	2 (5)	61 (10–113)		224 (91–356)	

PFS = progression-free survival, OS = overall survival, CTC-I = circulating tumor cells index, RECIST-I = radiological response evaluation index, PI = progression index, CHT = chemotherapy, HT = hormone therapy, TT = target therapy.

## Data Availability

Data will be available on reasonable request.

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
