# Peer review of "Early Detection of Disease Progression in Metastatic Cancers: Could CTCs Improve RECIST Criteria?"

_biomedicines, 2024, doi:10.3390/biomedicines12020388_

Round 1

Reviewer 1 Report

Comments and Suggestions for Authors

Valentina et al. concentrate on the early identification of disease progression in patients with metastatic tumors, specifically in those with prostate, colorectal, and breast cancers. The researchers assess whether circulating tumor cells (CTCs) could be incorporated into hybrid criteria to enhance the current radiologic evaluations (RECIST) that are employed in therapy selection.

Overall, the results presented in this paper are acceptable. However, I do suggest that the authors rewrite the paper with the majority. Here are some important points to remember:

1.     It's unclear how sampling time points are described. The patient's blood sample was collected concurrently with the radiograph, as far as I can figure out from the manuscript (line 115). It is unclear, nevertheless, how long it will take to get from the moment of sample to the relevant treatment period. This is a crucial period of time. Given that the authors noted that various patients get various treatment options, various treatment programs should call for varying radiographic examination intervals. In addition, the medications employed in various therapy modalities vary. Additionally, the authors did not explain how patients' treatment options varied from one another. It is advised that the authors include pertinent data to see whether variations in CTC counts under various drug regimens also serve as independent risk factors for the course of the disease.

2.     The Materials and Methods section did not contain a clear prohibition on CTC counting from the authors. The author's CTC criteria for prostate and breast cancer are obvious from Materials and Methods 2.2: "the increase in more than 5 CTCs per 7.5 ml of circulating blood is considered to be disease progression." PMID17289886 study findings indicate that in about 30–35% of metastatic breast tumors, CTCs are not pathologically detectable. Without a baseline, it is impossible to assess the threshold's reliability when the number of CTCs is low. A result of the possibility that the creteira is already 100% or 200% of the starting point. Furthermore, why are there various CTCs-I of different tumor type? Is it attributed to various tumor types? What is the groundwork?  If CTCs-I varies depending on the type of tumor, is it reasonable to assume that CTCs-I is a tumor-specific parameter? Then further reasoning: can patients of diverse ages and genders not be able to employ a fixed CTCs-I? CTCs may be impacted by variables such the type of tumor, patient age, gender, etc. Lastly, during the collection of patient blood samples, are there duplicate samples? Owing to the tiny number of CTCs, random factors will impede a single sampling. If there is no repeat sampling for the same patient, the outcome of one sampling may differ significantly from the true value, leading to unreliable results. Despite the fact that there is data from several people—as was already mentioned—different patients will have vastly diverse treatment regimens, as well as differences in gender, age, and other characteristics.

3.     The introduction's lines 73–74 state that because RECIST omits the biological process of tumor clonal proliferation, it will overestimate the rate at which the disease progresses. Please provide the author's explanation for this statement. My belief is that clonal expansion frequently leads to tumor development, particularly rapid growth following treatment. Typically, drug-resistant clones exhibit such rapid expansion, or clonal proliferation. The progression of the disease is closely linked to this phenomena. Why is disease progression overstated if RECIST shows fast tumor growth? It ought to be a typical and accurate estimation.

Furthermore, a few brief suggestions and questions are listed below.

·      Regulating the use of punctuation is necessary. The end of line 84 does not have a period. The periods on lines 349 and 362 are repeated twice. Line 97 starts with an extra space at the beginning, etc.

Comments on the Quality of English Language

Regulating the use of punctuation is necessary. The end of line 84 does not have a period. The periods on lines 349 and 362 are repeated twice. Line 97 starts with an extra space at the beginning, etc.

Author Response

REVIEWER #1

Valentina et al. concentrate on the early identification of disease progression in patients with metastatic tumors, specifically in those with prostate, colorectal, and breast cancers. The researchers assess whether circulating tumor cells (CTCs) could be incorporated into hybrid criteria to enhance the current radiologic evaluations (RECIST) that are employed in therapy selection.

Overall, the results presented in this paper are acceptable. However, I do suggest that the authors rewrite the paper with the majority. Here are some important points to remember: 

Q1: It's unclear how sampling time points are described. The patient's blood sample was collected concurrently with the radiograph, as far as I can figure out from the manuscript (line 115). It is unclear, nevertheless, how long it will take to get from the moment of sample to the relevant treatment period. This is a crucial period of time. Given that the authors noted that various patients get various treatment options, various treatment programs should call for varying radiographic examination intervals. In addition, the medications employed in various therapy modalities vary.

A1:  The blood sample is obtained at the same time of the radiological/clinical evaluation. As suggested by the good clinical practice guidelines, the cancer treatment is started in minimum 3, maximum 7 days. Disease evaluation were scheduled at three-months intervals.

Q2: Additionally, the authors did not explain how patients' treatment options varied from one another. It is advised that the authors include pertinent data to see whether variations in CTC counts under various drug regimens also serve as independent risk factors for the course of the disease.

A2: We thank the reviewer for the comment. We carried out an analysis according to the treatment options (chemotherapy, hormone therapy, biological agent) and the CTCs counts (mean value and standard deviation) do not present significant differences between the groups. A related comment is introduced in the revised version of the manuscript.

Q3.     The Materials and Methods section did not contain a clear prohibition on CTC counting from the authors. The author's CTC criteria for prostate and breast cancer are obvious from Materials and Methods 2.2: "the increase in more than 5 CTCs per 7.5 ml of circulating blood is considered to be disease progression." PMID17289886 study findings indicate that in about 30–35% of metastatic breast tumors, CTCs are not pathologically detectable. Without a baseline, it is impossible to assess the threshold's reliability when the number of CTCs is low. A result of the possibility that the creteira is already 100% or 200% of the starting point. Furthermore, why are there various CTCs-I of different tumor type? Is it attributed to various tumor types? What is the groundwork?  If CTCs-I varies depending on the type of tumor, is it reasonable to assume that CTCs-I is a tumor-specific parameter?

A3. The cutoff values for CTCs vary according to the type of cancer, with a set limit of 5 CTCs/7.5ml for breast and prostate cancers, and 3 CTCs/7.5ml for colorectal cancer. These tumor-specific cutoffs have been established based on literature references (Ref.s 10, 11, and 12). Moreover, as reported in Table 1 and now also in the text, the criteria for an increase in the number of CTCs take into account the baseline condition. In particular, if the baseline CTC count exceeds the specific cutoff, any subsequent rise in CTCs is assumed significant.

Q5: Then further reasoning: can patients of diverse ages and genders not be able to employ a fixed CTCs-I? CTCs may be impacted by variables such the type of tumor, patient age, gender, etc.

A5: Thank for the interesting comment. As previously discussed, the cutoff values for CTCs are tumor specific. The pertinent literature does not report, to the best of the authors knowledge, evidence of a possible influence of  CTCs count by gender or age of patients.

Q6: Lastly, during the collection of patient blood samples, are there duplicate samples? Owing to the tiny number of CTCs, random factors will impede a single sampling. If there is no repeat sampling for the same patient, the outcome of one sampling may differ significantly from the true value, leading to unreliable results. Despite the fact that there is data from several people—as was already mentioned—different patients will have vastly diverse treatment regimens, as well as differences in gender, age, and other characteristics.

A6: Thank you for the interesting comment. We observe that the international prospective trials that led to the FDA approval of the CellSearch@ system and the corresponding reference values for CTC enumeration, did not involve the triple-copy repetition of sampling. In particular in the study referred above (PMID17289886) and now introduced in the references list, a comparison between the analysis of identical samples in different centers reported no differences on the CTC counts.

Q7: The introduction's lines 73–74 state that because RECIST omits the biological process of tumor clonal proliferation, it will overestimate the rate at which the disease progresses. Please provide the author's explanation for this statement. My belief is that clonal expansion frequently leads to tumor development, particularly rapid growth following treatment. Typically, drug-resistant clones exhibit such rapid expansion, or clonal proliferation. The progression of the disease is closely linked to this phenomena. Why is disease progression overstated if RECIST shows fast tumor growth? It ought to be a typical and accurate estimation.

A7 Thank you for the comment. We agree with the reviewer observation. The typo was fixed, and a better description is now added.

Q8 Furthermore, a few brief suggestions and questions are listed below.

Regulating the use of punctuation is necessary. The end of line 84 does not have a period. The periods on lines 349 and 362 are repeated twice. Line 97 starts with an extra space at the beginning, etc.

A8 We corrected text as requested.

Reviewer 2 Report

Comments and Suggestions for Authors

This manuscript authored by Valentina Magri and colleagues titled “Early detection of disease progression in metastatic cancers: could CTCs improve RECIST criteria?” is interesting and trying address an important issue in metastatic cancer treatment. Manuscript is well written, easy to read and experimental design is acceptable. There are some minor issues authors need to address before accepting for publication.

Abstract

1.       Lines 24 & 25 define PES and OS since this is the first time they appear in the manuscript.

2.       Line 27 …..than RECIST-I the progression disease…… should be changed to than RECIST-I the disease progression …….

Materials and Methods

2.1   CTC enumeration

1.       Does your institution has a CellSearch® system? Or did you send samples to somewhere else? Please mention in this section where you did CTC enumeration using CellSearch system.

Discussion

Authors should discuss the utility of cell-free tumor DNA for early detection of disease progression. Has anybody tried to use cell-free tumor DNA to improve RECIST criteria? Please discuss these items in your discussion section to improve the quality of your manuscript.

Author Response

REVIEWER #2

This manuscript authored by Valentina Magri and colleagues titled “Early detection of disease progression in metastatic cancers: could CTCs improve RECIST criteria?” is interesting and trying address an important issue in metastatic cancer treatment. Manuscript is well written, easy to read and experimental design is acceptable. There are some minor issues authors need to address before accepting for publication.

Q1: Abstract

  1. Lines 24 & 25 define PES and OS since this is the first time they appear in the manuscript.
  2. Line 27 …..than RECIST-I the progression disease…… should be changed to than RECIST-I the disease progression …….

A1: Thank you for the comments. We corrected text as requested.

Q2: 2.1   CTC enumeration

 Does your institution has a CellSearch® system? Or did you send samples to somewhere else? Please mention in this section where you did CTC enumeration using CellSearch system.

A2: The CTC enumeration was carried out with the CellSearch® device (Menarini silicon biosystems), which is a part of our group's equipment and is available in the Liquid Biopsy Unit within the Department of Molecular Medicine since 2010.  The text was modified accordingly.

Q3 Discussion: Authors should discuss the utility of cell-free tumor DNA for early detection of disease progression. Has anybody tried to use cell-free tumor DNA to improve RECIST criteria? Please discuss these items in your discussion section to improve the quality of your manuscript.

A3: Thank you for the suggestion, we discussed the utility of cell-free tumor DNA for early detection of disease progression and insert it in discussion section.

 In order to improve RECIST criteria, the use of circulating tumor DNA (ctDNA) is also currently under consideration. Occurring early after the beginning of treatment, ctDNA variations can potentially predict response to therapy weeks before any evidence of changes in tumor size [a]. Gouda et al. (2022) recently reported the analysis of serial blood samples collected at baseline, mid-

treatment and at the time of restaging from patients with different tumor types [b]. The results of this study highlight that higher ctDNA detection rates can be observed in patients with disease progression at first restaging compared to patients with stable disease, partial response, or complete response [b]. Despite limitations, the authors reported an association between detection of ctDNA and response to therapy, supporting the potential of ctDNA analysis as a valid alternative to

radiologic imaging in evaluating treatment outcomes [b]. Although growing evidence seems to confirm the role of ctDNA in the assessment of treatment response, the absence of methodological standardization, together with the lack of a globally accepted definition of ctDNA response and progression critically hamper the implementation of ctDNA response criteria in the clinical practice

[a]. Further investigations are also needed to better define which source is more suitable for ctDNA analysis, the clinical setting for this application and the timing of the assessment [a].

References:

[a] Gouda, M.A., Janku, F., Wahida, A., Buschhorn, L., Schneeweiss, A., Karim, N.A., Perez, D.D.M., Del Re, M., Russo, A., Curigliano, G. and Rolfo, C., 2023. Liquid Biopsy Response Evaluation Criteria in Solid Tumors (LB-RECIST). Annals of Oncology.

[b] Gouda, M.A., Huang, H.J., Piha-Paul, S.A., Call, S.G., Karp, D.D., Fu, S., Naing, A., Subbiah, V., Pant, S., Dustin, D.J. and Tsimberidou, A.M., 2022. Longitudinal monitoring of circulating tumor DNA to predict treatment outcomes in advanced cancers. JCO Precision Oncology, 6,p.e2100512.

Reviewer 3 Report

Comments and Suggestions for Authors

the manuscript should focused on only breast cancer patients, in this form it is too multi-variable and too under-powered to achieve any conclusion

Author Response

REVIEWER #3

The manuscript should focused on only breast cancer patients, in this form it is too multi-variable and too under-powered to achieve any conclusion.

Answer: We thank the reviewer for the comment. Even if we partially agree with the reviewer’s point of view, we decided to maintain the complete analysis. Though in this analysis significant results were obtained only in patients affected by breast carcinoma, the CellSearch system is FDA approved for the three tumor types: breast, colorectal and prostate cancer. In view of future suggested trials aimed at integrating RECIST and CTCs count in new criteria, all the three cancers should be investigated and, therefore, we considered necessary to report all the recorded data systematically conducted.

Round 2

Reviewer 1 Report

Comments and Suggestions for Authors

Although the author provided point-to-point answers to my questions, some questions still did not satisfy me.

Q1: Regarding A1, the author did not provide answers to key questions. This answer is simply a restatement of the description in the Materials and Methods section of the manuscript regarding blood sampling. My core question is that different treatment strategies will have different effects on the changes in CTCs. The fluctuation of CTC content after treatment is currently unknown. Assuming that treatment is effective, the number of CTCs will decrease. Errors will be introduced into the numerical statistics of CTCs if patients with different treatment strategies are sampled at the same time interval following treatment. Because at this sampling time point, the number of CTCs in samples with different treatment strategies is declining at different rates. The authors should specify how long it takes to collect a sample of blood under various treatment regimens and how long it takes to get the most recent treatment. It is required that the period between the most recent treatments has been sampled and the treatment regimens that the sample has received are shown in tabular form. I believe the authors neglected to consider the effect of various treatment techniques on CTCs if the time interval between the sample's most recent treatment and the time of blood sample collection for each treatment strategy is the same. The author also presented two new concepts in this response: Disease evaluation and radiological/clinical evaluation. What distinguishes these two concepts from one another?

Q2: Regarding A2, the author did not include relevant clinical information about the sample in the revised manuscript. In addition, the authors stated that they performed correlation analyses to demonstrate that CTC counts did not differ significantly between groups. However, the relevant analysis results are not displayed, only a simple sentence description.

Q3: Regarding A6, there is no difference between CTC statistics performed on the same sample by different institutions. This can only prove the stability of the CTC counting method and has nothing to do with sampling. The manuscript does not address how much sample size should be collected at one time during blood sampling. Based on the loading requirements of CellSearch, at least 7.5 mL of peripheral blood is required. Compared with taking only 8 ml or less of blood sample at a time, collecting 8 ml of blood in batches multiple times can better ensure the accuracy of CTC counting. The authors did not set up replicate samples of peripheral blood. The impact of this should be discussed in the discussion section.

Round 2

Reviewer #1

Q1: Regarding A1, the author did not provide answers to key questions. This answer is simply a restatement of the description in the Materials and Methods section of the manuscript regarding blood sampling. My core question is that different treatment strategies will have different effects on the changes in CTCs. The fluctuation of CTC content after treatment is currently unknown. Assuming that treatment is effective, the number of CTCs will decrease. Errors will be introduced into the numerical statistics of CTCs if patients with different treatment strategies are sampled at the same time interval following treatment. Because at this sampling time point, the number of CTCs in samples with different treatment strategies is declining at different rates. The authors should specify how long it takes to collect a sample of blood under various treatment regimens and how long it takes to get the most recent treatment. It is required that the period between the most recent treatments has been sampled and the treatment regimens that the sample has received are shown in tabular form. I believe the authors neglected to consider the effect of various treatment techniques on CTCs if the time interval between the sample's most recent treatment and the time of blood sample collection for each treatment strategy is the same. The author also presented two new concepts in this response: Disease evaluation and radiological/clinical evaluation. What distinguishes these two concepts from one another?

A1: We apologize for not having fulfilled the request in the previous response. The present study is retrospective and patients were sampled and analysed (within 2 days), during the disease evaluation (which we consider as synonym of clinical/radiological evaluation) approximately every 3 months or at progression disease, regardless of the treatment. Our analysis was conducted based on previous research experiences adopted as reference for the liquid biopsy. In a pooled analysis, conducted on 1944 eligible patients and published on Lancet Oncology on 2014, Bidard et al confirm the independent prognostic significance of CTC. Furthermore, they demonstrated that CTC enumeration enhances prognostic accuracy in metastatic breast cancer when integrated into comprehensive clinicopathological predictive models. In this comprehensive analysis, patients included underwent various types of treatment modalities, including chemotherapy, hormone therapy, chemotherapy combined with anti-HER2 targeted therapy, chemotherapy combined with Bevacizumab or other targeted therapies. However, blood sampling was consistently conducted at the same intervals (baseline, 3-4 weeks, and 6-8 weeks) regardless of the specific treatment regimen administered. While this analysis demonstrated the robust validity of circulating tumor cell enumeration at predefined time points (baseline, 3-4 weeks, and 6-8 weeks), it did not specifically address the potential impact of different treatment modalities on CTC sampling and counts. All studies conducted utilizing the CellSearch System for CTC enumeration are affected by the same potential bias, based on the best of our knowledge. Nevertheless, we agree with the Reviewer's suggestion that a tailored analysis focusing on specific treatment modalities, particularly in relation to sampling strategies, warrants further investigation. We have extensively addressed this topic in the manuscript.

Q2: Regarding A2, the author did not include relevant clinical information about the sample in the revised manuscript. In addition, the authors stated that they performed correlation analyses to demonstrate that CTC counts did not differ significantly between groups. However, the relevant analysis results are not displayed, only a simple sentence description.

A2: We observe that, according to previous published studies [Cristofanilli2006] the detectability of CTC seems not related to the specific therapy, but we agree that the number of CTCs count can be affected by the particular treatment. In others terms the underlying characteristic of the tumor which leads to different drugs protocols could be mainly responsible of the differences of CTCs counts, now introduced in the manuscript. In our study, the cohort of patients affected by colorectal and prostate cancer were treated with a single therapy (chemotherapy+ bevacizumab in colorectal and endocrine therapy in prostate), we reported the results for breast cancer, where different treatments were adopted (new Table 6). We discussed the relevant difference and provided a possible justification.

Moreover, the effect of therapy in the breast cancer patients was considered also with reference to the progression index on the PFS and OS. A new Table 7 has been introduced.

Q3: Regarding A6, there is no difference between CTC statistics performed on the same sample by different institutions. This can only prove the stability of the CTC counting method and has nothing to do with sampling. The manuscript does not address how much sample size should be collected at one time during blood sampling. Based on the loading requirements of CellSearch, at least 7.5 mL of peripheral blood is required. Compared with taking only 8 ml or less of blood sample at a time, collecting 8 ml of blood in batches multiple times can better ensure the accuracy of CTC counting. The authors did not set up replicate samples of peripheral blood. The impact of this should be discussed in the discussion section.

Q3: Thank you for your comment. In our research, we conducted a single withdrawal of 7.5 ml from each patient in the morning on the day of each disease (clinical/radiological) evaluation. All enrolled patients were day hospital attendees and not under inpatient care. In a day hospital setting, patients left the structure at the end of the treatment, making almost impossible to split the sampling into a multiple set of withdrawals.

To the best of our knowledge and based on current literature, there are no studies using samples collected at different times. We appreciate your suggestion as many pieces of evidence indicate that the circadian rhythm may play a significant role in the entire process of tumorigenesis (Huang2023) and has the potential to be applied in detecting circulating tumor cells (CTCs). It can be hypothesized that samples collected at different times might monitor better the biological status of the disease compared to a single morning withdrawal. From a scientific standpoint, collecting blood for the detection of circulating tumor cells (CTCs) at different times of the day has a meaningful rationale, and we propose to be considered in future trials. This approach could provide valuable insights into the temporal dynamics of CTCs and enhance our understanding of the disease. We discussed the matter manuscript.

Bidard FC, Peeters DJ, Fehm Tet al. Clinical validity of circulating tumour cells in patients with metastatic breast cancer: a pooled analysis of individual patient data. Lancet Oncol. 2014 Apr;15(4):406-14. doi: 10.1016/S1470-2045(14)70069-5. Epub 2014 Mar 11. PMID: 24636208.

Cristofanilli, M. Circulating Tumor Cells, Disease Progression, and Survival in Metastatic Breast Cancer. Semin. Oncol. 2006, 33(SUPPL. 9), 9–14. https://doi.org/10.1053/j.seminoncol.2006.03.016.

Huang, C.; Zhang, C.; Cao, Y.; Li, J.; Bi, F. Major Roles of the Circadian Clock in Cancer. Cancer Biol. Med. 2023, 20 (1), 1–24. https://doi.org/10.20892/j.issn.2095-3941.2022.0474.

Reviewer 3 Report

Comments and Suggestions for Authors

the authors hasn`t take into consideration the recommendations of this reviewer

Round 2

Reviewer #3

The manuscript should focused on only breast cancer patients, in this form it is too multi-variable and too under-powered to achieve any conclusion.

Answer: we emphasize our regret for not having responded comprehensively. We underscore the significance of investigating all three tumor types—breast, colorectal, and prostate cancer—given the FDA approval of the CellSearch system for these cancers. Consequently, we deemed it necessary to systematically report all recorded data. Moreover, considering the scarcity of documented similar experiences in the scientific literature, we believe it is crucial to publish our entire experience, including results that did not reach statistical significance.

Our intention is to provide a foundational experience for future studies focusing on the same topic, offering valuable guidance in study design despite being underpowered, particularly concerning results related to prostate and colorectal carcinoma. We also advocate for the potential introduction of circulating tumor cells (CTCs) into clinical practice as a valuable tool for reassessing tumor pathology alongside the well-established RECIST criteria. However, we acknowledge that robust evidence will be essential in the near future to support its integration into everyday clinical practice.

In conclusion, we assert the importance of publishing the entire analysis, encompassing not only the results from the patient cohort with breast carcinoma, where statistical significance was achieved, but also the results from patients with colorectal and prostate carcinoma. Despite not reaching statistical significance due to discussed limitations in the manuscript, we believe sharing these findings contributes to a more comprehensive understanding of our study and could prove valuable for future research in these specific patient populations.

Round 3

Reviewer 1 Report

Comments and Suggestions for Authors

The authors gave a comprehensive response to my questions. Related issues have also been revised in the revised manuscript. I believe that the current version meets the requirements for publication.

Reviewer 3 Report

Comments and Suggestions for Authors

The authors hasn`t answered this reviewer`s concerns